# Pulsed Electric Field-Assisted Ethanolic Extraction of Date Palm Fruits: Bioactive Compounds, Antioxidant Activity and Physicochemical Properties

**Azhari Siddeeg [1], Muhammad Faisal Manzoor [2,3], Muhammad Haseeb Ahmad [4], Nazir Ahmad [4,*], Zahoor Ahmed [2,3], Muhammad Kashif Iqbal Khan [5], Abid Aslam Maan [5], Mahr-Un-Nisa [4], Xin-An Zeng [2,3] and Al-Farga Ammar [6]**

[1]  Department of Food Engineering and Technology, Faculty of Engineering and Technology, University of Gezira, Wad Medani 21111, Sudan

[2]  School of Food Science and Engineering, South China University of Technology, Guangzhou 510640, China

[3]  Overseas Expertise Introduction Center for Discipline Innovation of Food Nutrition and Human Health (111 Center), Guangzhou 510640, China

[4]  Institute of Home and Food Sciences, Faculty of Life Science, Government College University, Faisalabad 38000, Pakistan

[5]  Department of Food Engineering, Faculty of Agricultural Engineering, University of Agriculture, Faisalabad 38000, Pakistan

[6]  College of Sciences, Biochemistry Department, University of Jeddah, Jeddah 21959, Saudi

*  Correspondence: drnazirahmad@gcuf.edu.pk

**Abstract:** The current study was aimed to observe the influence of pulsed electric field (PEF) on the extraction of bioactive components; antioxidant activity and physicochemical properties of date palm fruit extract (DPFE) as compared to the extract untreated by PEF. The fruit was treated with PEF (frequency: 10 Hz, time: 100 µs, pulses number: 30, electric field strength (EFS): 1, 2, and 3 kV/cm. The results show that PEF has a positive impact on the total content of carotenoids, anthocyanins, flavonoids, and phenolics by increasing the EFS. DPFE treated with PEF exhibit a strong antioxidant activity as compared to untreated extract, while electrical conductivity, pH values, and titratable acidity were not affected by PEF. The results indicate a notable increase in the volatile components in DPFE treated with PEF at 3 kV/cm. Thus, PEF treatment can assist the ethanolic extraction of DPFE to improve the bioactivity and antioxidative activity. These findings suggest that PEF can be a more suitable technique to enhance solvent extraction on a commercial basis.

**Keywords:** dates; PEF; bioactive compounds; antioxidant activity; ethanolic extraction

## 1. Introduction

The date palm (Sukkari variety) is a famous variety grown in the Kingdom of Saudi Arabia and Iraq because of its high economic revenues to buyers and farmers for its high quality [1]. These fruits are good for human health, especially for those with poor heart conditions and also suitable for the digestive process due to their high fiber content [2]. The date fruit is considered a source of antioxidant, phenolic components, anti-mutagenic activity and medicinal values [3]. In previous studies, proximate analyses, the antioxidative and chemical composition of many kinds of date palm have been reported [4,5]. They are rich in sugar and minerals [6], with a low level of lipids and proteins. Pulsed electric field (PEF) is considered an alternative to thermal production of food products. One of the key benefits assisted with these techniques is to promote the nutritional value because of minimum thermal dilapidation [7,8]. PEF as a technique can generate permeabilization of cell membranes

when they are treated with short treatment time and low energy consumption to minimize quality deterioration of food compounds [9–11]. The electroporation formation in the cell membrane is the basic principle of PEF technology that leads to a high extraction yield. Pulse electric force after passing from cell membrane creates charge on molecules, which assist the molecules to separate on charge mass basis [12]. The impacts of high-intensity of PEF processing on physicochemical and antioxidant characteristics were assessed in date palm juice which stored at refrigeration temperature. The high intensity of PEF contributes to preserve the juice during storage with good quality, which is better than heat treatment [13]. Therefore, PEF has garnered the attention for it industrial application due to beingnon-thermal, cost-effective, simple, constant and efficient for bioactive molecule extraction [14,15]. Food industries are paying attention to the novel techniques that can preserve the juices and liquid food products in natural forms for longer storage time with least nutrients losses [16]. In PEF technique, the cell is exposed to PEF temporarily, which can encourage the destabilization of lipid, proteins and bilayer in cell membranes [17]. Many studies have been reported the impact of PEF on chemical composition and quantitative analysis of an extract from different foods [18–20]. PEF is an emerging technique and has better results for the extraction of bioactive compounds from onion [21], orange peel [22], grape juices [23] and *Borago officinalis* leaves [24].

Water and ethanol blends are generally used for extraction from plant materials due to high solubility for a wide range of phenolics. The solvent mixtures of water and ethanol are not toxic for humans [4]. In our previous study, we have shown that ethanol was better extraction solvent than methanol for date palm fruit [25]. Thus, in this study, we have used the ethanol-assisted PEF extraction and characterization of bioactive molecules, as no study has been done on phenolic compounds, antioxidant activity, color attributes and volatile components of date palm fruit extract though PEF technology according to our knowledge. Therefore, PEF treatment assisted ethanolic extraction of date palm fruits has been presented in this study.

## 2. Materials and Methods

### 2.1. Material and Chemicals

The fresh date palm (*Phoenix dactylifera* L., Sukkari variety) was purchased from the market of Guangzhou city and used in the laboratory. The fruit was cleaned by separating damaged fruit and stored at ambient temperature until further analysis. All chemical and reagents were brought from Aladdin, otherwise mentioned.

### 2.2. PEF Treatment

Lab-scale PEF (SCUT-PEF) was used for extraction from 100 g of date palm fruit flesh. The operating conditions of PEF were set between two parallel chambers of dimensions 6.55 and 4.36 cm. The 1, 2 and 3 kV/cm electric field strengths with 30 μs pulse number were applied for 100 μs period at 10 Hz frequency. The PEF application was made possible so that sample temperature did not exceed above 5 °C. Equation 1 was used to calculate specific energy (*W*) accordingly [26]:

$$W = \frac{0.5 \times V^2 \times C \times n}{m} \tag{1}$$

### 2.3. Ethanolic Extraction

Ethanolic extraction was carried out using following procedure. The 100 g fruit was used for extraction in an orbital shaker using a mixture of 300 mL ethanol-water at the ratio of 4:1 (v/v) for 6 h. The extract was filtered and centrifuged at 4000 rpm for 10 min. The extract was vacuum condensed at 45 °C for 3 h to obtain final date palm fruit extract (DPFE). The DPFE was stored in dark glass flasks at 4 °C in a freezer until use. Finally, the supernatant of DPFE was lyophilized using a freeze-dryer to calculate the total phenolic content (TPC) of DPFE. A Folin–Ciocalteu reagent was used to determine

the TPC using following procedure [27]. DPFE (200 mg) was mixed with 1.0 mL of Folin–Ciocalteu reagent and vertexed gently for 2 min. A 3.0 mL volume of $Na_2CO_3$ was poured in the sample tube and vortexed carefully. Furthermore, the volume was made with20 mL ofdistilled water and at 50 °C for 20 min. Moreover, the sample was subjected to centrifugation (3000 rpm) for 10 min and UV-spectrophotometer absorbance was recorded at 750 nm. Gallic acid equivalent (GAE) was used as standard for linear equation and TPC of DPFE was calculated accordingly.

## 2.4. Determination of Total Flavonoids Content

The procedure descried earlier was used to determine total flavonoids content (TFC) [10]. A 2.0 mL of DPFE was mixed with concentrated 5% $NaNO_2$ (600 μL) and 10% $AlCl_3$ (600 μL). A 4.0 mL of NaOH was added in test tube and incubated at 24 °C for 5 min. After incubation, distilled water was added to make 20 mL of volume and solution was prudently vortexed. UV-Spectrophotometer absorbance was measured at 510 nm. The values of TFC were calculated as catechin equivalents (CE: mg/100 g).

## 2.5. Determination of Total Carotenoids Content

A previously described procedure was used to determined total carotenoids content (TCC) with slight modification [28]. DPFE (50 mL) was mixed with 160 mL of mixture of n-hexane and acetone. The aqueous stage was continually extracted after color disappearance. The sample was dehydrated by addition of sodium sulfate. Standard curve was obtained using absorbance (450 nm) of different β-carotene dilution as standard. The absorbance of sample was measured using spectrophotometer (450 nm) at ambient temperature. The concentration of TCC was calculated as μg of β-carotene equivalent/mL of date extract.

## 2.6. Determination of Total Anthocyanins Content

Total anthocyanins content of DPFE (*TAC*) was determined using previously reported method with little modification [29]. The DPFE (2 mL) was blended with 18 mL of buffer solution system. The buffer solution system was prepared from 0.025 M potassium chloride (pH 1.0) and 0.4 M, sodium acetate (pH 4.5) buffers. The absorbance was measured at 520 and 700 nm, using a spectrophotometer. *TAC* was calculated by Equation (2):

$$TAC(\text{mg/l}) = \frac{Abs \times molecular\ weight \times dilution\ factor \times 1000}{\varepsilon \times path\ length\ (1\ cm)} \tag{2}$$

where "$\varepsilon$" is the extinction coefficient (28,000 l/mol/cm).

## 2.7. Antioxidant Activity Measurement

### 2.7.1. Reducing Power Assay

The reducing power assay of DPFE and the untreated extract was carried out accordingly [30]. Different concentration (150, 300, 450 and 600 μL/mL) of DPFE were blended with 10 mg/mL of potassium ferricyanide and 200 μL sodium phosphate and incubated for 30 min at 50 °C. A volume (680 μL) of the reaction solution was blended with 680 μL of distilled water and 68 μL of ferric chloride (to make a concentration of 10 mg/mL. The absorbance was measured at 700 nm. A 0.3 mM of vitamin C was used as standard.

### 2.7.2. DPPH Radical Scavenging Activity

The 2,2-diphenyl picrylhydrazyl (DPPH) of DPFE was determined using a procedure reported by Oliveira, et al. [31], with a slight modification. Different concentrations of DPFE (150, 300, 450 and 600 μL/mL) in the solvent were blended with 3.5 mL of DPPH solution and incubated for 30 min at

25 °C. The absorbance values were taken at 517 nm using a spectrophotometer. Equation (3) was used to calculate the DPPH activity (%):

$$DPPH\ \% = (A_C - A_S/A_C) \times 100 \tag{3}$$

where "$A_S$" is absorbance of *DPFE* and "$A_C$" was absorbance of control, vitamin C was used as standard.

### 2.8. Measurement of Physicochemical Characteristics of Date Palm Fruit Extracts

The titratable acidity (TA), total soluble solids (TSS), pH and electrical conductivity (EC) of DPFE and untreated extract were determined. The TSS (°Brix) was measured through an Abbe refractometer. The pH values were measured using a pH-meter; EC was measured with a conductivity meter. TA was measured according to the procedure reported by Aadil et al. [16]. Colorimeter (CR-400) was used for color measurement. The hue angle ($h°$) was calculated using following Equation (4):

$$h° = tan/b*/a* \tag{4}$$

The parameters of color: the difference between green and red color ($a*$); the difference between blue and yellow color ($b*$); luminosity ($L*$) were determined, and total color variation was calculated using following Equations (5) and (6):

$$\Delta E = \sqrt{(\Delta a*)^2 + (\Delta b*)^2 + (\Delta L*)^2} \tag{5}$$

$$\Delta L* = \Delta L_{sample} - \Delta L_{control};\ \Delta a* = a_{sample} - a_{control};\ \Delta b* = b_{sample} - b_{control} \tag{6}$$

Chroma value ($C*$) was calculated by Equation (7), as reported by Saricoban and Yilmaz [32]:

$$C* = \sqrt{a^2 + b^2} \tag{7}$$

### 2.9. Determination of 5-(Hydroxymethyl) Furfural (5-HMF)

The concentration of 5-HMF was measured by the procedure reported by Mtaoua et al. [13]. Different concentrations (2–8 mg/L) of HMF were prepared to plot the calibration curve and to calculate the HMF concentration.

### 2.10. Determination of Volatile Flavors Compounds

The volatile flavor contents of DPFE were determined using gas chromatography-mass spectrometry (Mass Hunter GC/MS Acquisition, Agilent 5977A Series, Folsom, CA 95630, USA). A volume of 8.0 mL of DPFE was poured in 2.5 mL vial, and 1.2 μL cyclohexanone (internal standard) was added. The extract was saturated with NaCl (50 mg), after equilibrating for 10 min at 45 °C, this sample was extracted at 45 °C for 40 min at a constant temperature with stirring in adaptable analyst with the solid-phase microextraction (SPME) capability. The sample was injected on HP-5 column (15 mm × 250 μm × 0.25 μm) with the help of helium gas at a flow rate of 1.5 mL/min. The oven was automated from 120 °C to 250 °C with a rate of 5 °C/min and maintained for 5 min. The volatile components were identified with the help of MS of the standard.

### 2.11. Statistical Analysis

SPSS version 16.0 for Windows (Chicago, IL, USA) was used to analyze data. The least significant test using analysis of variance (ANOVA) was used for significant differences and Duncan's test was used for multiple comparisons between means at a significance level of $p < 0.05$.

## 3. Results and Discussion

### 3.1. Total Phenolic (TPC) and Flavonoids Contents (TFC)

Table 1 shows the results for the TPC and TFC of DPFE treated by PEF to explore a significant difference. TPC of treatments DPFE1, DPFE2, and DPFE3 was 64.20, 65.90 and 67.35 mg GAE/100 g, respectively, as compared with untreated extract (62.50 mg GAE/100 g). The highest value for TPC was observed in DPFE3 which was significantly different ($p < 0.05$) as compared to others treatments. This difference might be due permeabilization of plant cells by PEF application to yield more fruit extracts production, increase the extractability and enhance the intracellular metabolites extraction [33].

**Table 1.** TPC and TFC of DPFE treated by PEF compared with the untreated extract.

| Sample | TPC (mg GAE/100 g) | TFC (mg CE/100 g) |
|---|---|---|
| Untreated extract | 62.50 ± 0.11 [d] | 3.20 ± 0.09 [d] |
| DPFE1 | 64.20 ± 0.41 [c] | 4.58 ± 0.71 [c] |
| DPFE2 | 65.90 ± 0.91 [b] | 5.80 ± 0.63 [b] |
| DPFE3 | 67.35 ± 0.71 [a] | 6.75 ± 0.55 [a] |

TPC: toal phenolic content, TFC: total flavonoids content, DPFE: date palm fruit extract, PEF: pulse electric field. DPFE1: 1 kV/cm, DPFE2: 2 kV/cm, DPFE3: 3 kV/cm. The tests were performed as triplicates and values are mean ± standard deviation. Different superscript alphabets in a row significant difference ($p < 0.05$) using Duncan's multiple range tests.

The presence of some enzymes can affect the TPC which are responsible for phenolic biosynthesis, and the induction of these enzymes activity can increase the accumulation of TPC [34,35]. As given in Table 1, the TFC results showed a significant increase in the TFC of DPFE (DPFE3 > DPFE2 > DPFE1) as compared to untreated extract. The TFC was increased from 3.20 mg CE/100 g (untreated extract) to 4.58 mg CE/100 g inDPFE1, 5.80 mg CE/100 g in DPFE2 and 6.75 mg CE/100 g in DPFE3. The same trend was found in some studies which have indicated a significant increase in TFC in different fruits extracts and juices [13,36].Therefore, compared with conventional extraction methods, PEF is considered as a pre-treatment process for fruits under relative conditions such as low temperature and neutral pH value, can significantly abbreviate extraction time and decrease the use of solvents.

### 3.2. Impact on Total Anthocyanin Contents (TAC)

As given in Table 2, data concerning TAC revealed a significant increase ($p < 0.05$) in TAC of DPFE treated with PEF as compared to the untreated extract was observed. TAC of DPFE1, DPFE2 and DPFE3 was 0.94, 1.23 and 2.08 mg/L, respectively, as compared to the untreated extract (0.75 mg/L). The same trend (significant increase in TAC) was reported after PEF treatment using a high frequency with low pulse widths for strawberry juice [37]. As reported in previous studies, a considerable change in TAC of products treated by PEF technique was found which may be due to the cavitation's process that regulates different biological reacting chemicals and an increase in the disintegration of affected particles as well as the diffusion rates [38]. Effect of PEF on extraction yields of anthocyanin of potato using ethanol and water (96% and 48%) as extraction solvents were found to increase the TAC [39]. Anthocyanins retention dependents on polarity, pulsed frequency width and treatment time during the treatment by PEF. A TAC increase has been observed in grapefruit treated by PEF [16].

**Table 2.** Total Anthocyanin Contents (TAC) of DPFE treated by PEF compared with the untreated extract.

| Sample | TCC (μg/mL) | TAC (mg/L) |
|---|---|---|
| Untreated extract | 2.85 ± 0.12 [d] | 0.75 ± 0.09 [d] |
| DPFE1 | 3.29 ± 0.09 [c] | 0.94 ± 0.11 [c] |
| DPFE2 | 4.93 ± 0.07 [b] | 1.23 ± 0.03 [b] |

<div align="center">**Table 2.** *Cont.*</div>

| Sample | TCC (µg/mL) | TAC (mg/L) |
| --- | --- | --- |
| DPFE3 | 6.10 ± 0.10 [a] | 2.08 ± 0.09 [a] |

TCC: total carotenoids content, DPFE: date palm fruit extract, PEF: pulse electric field. DPFE1: 1 kV/cm, DPFE2: 2 kV/cm, DPFE3: 3 kV/cm. The tests were performed as triplicates and values are mean ± standard deviation. Different superscript alphabets in a row show significant difference ($p < 0.05$) using Duncan's multiple range tests.

### 3.3. Total Carotenoids Contents (TCC)

As illustrated in Table 2, the data concerning TCC in DPFE treated by PEF were compared with the untreated extract. A significant increase ($p < 0.05$) in TCC with PEF was observed. TCC was 3.29, 4.93 and 6.10 µg/mL for DPFE1, DPFE2 and DPFE3, respectively, as compared to untreated extract (2.85 µg/mL). The TCC increasing in DPFE treated by PEF can be due to ruptured cell walls within cavitation's process and as a result, free carotenoids release [40]. A considerable increase in TCC was noticed in orange-carrot juice treated with high intensity of PEF [41]. The trend of increase in our results was found similar as reported in some other studies using PEF treatment [16,42].

### 3.4. Antioxidant Activities

#### 3.4.1. DPPH Assay

Figure 1 shows the DPPH results of DPFE treated by PEF and the radical scavenging potential of the reference. According to data obtained, the DPPH results of DPFE1, DPFE2 and DPFE3 at different concentrations were in the range 45–66, 48–68 and 50–72%, respectively, and an inhibition concentration by 50% ($IC_{50}$) of 209.60 ($R^2 = 0.95$), 156.09 ($R^2 = 0.93$) and 110 ($R^2 = 0.95$) µL/mL, respectively. A higher radical scavenging activity in PEF treated extract was observed compared to the untreated extract; however, this was lower ($p < 0.05$) as compared to reference (vitamin C) at the same concentrations. The results are similar to the previous finding that PEF can increase free radical scavenging ability [43,44]. A significant increase was observed which can probably be due to improved extraction level of TCC, TPC, TAC and TFC as a result of cavitation processes after PEF treatment with different intensities. The strongest DPPH activity was noticed in DPFE3; this can be as a result of the synergistic impact of the highest PEF intensity (3 kV/cm) in that extract. The similar trend in activity was reported earlier studies used PEF treatment for liquid food products [40,45].

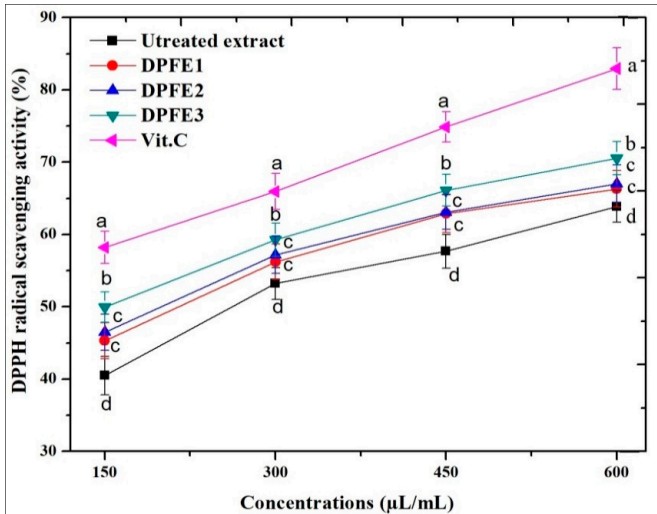

**Figure 1.** DPPH radical scavenging activity of DPFE treated by PEF as compared to untreated extract. DPFE1: 1 kV/cm, DPFE2: 2 kV/cm, DPFE3: 3 kV/cm. The tests were performed as triplicates and values are mean ± standard deviation. Different superscript alphabets in a row show significant difference ($p < 0.05$) using Duncan's multiple range tests.

### 3.4.2. Reducing Power Assay

As reported, reducing the power of the bioactive components is used to evaluate their aptitude to give an electron, and may help as an indicator of their antioxidant capacity [46]. The reducing powers of DPFE treated by PEF and untreated extract along with vitamin C as reference are presented in Figure 2. The absorbance of DPFE1, DPFE2 and DPFE3 was found to be in the range of 0.23–0.41, 0.33–0.54 and 0.41–0.65, respectively, which were found to be higher ($p < 0.05$) in the treated extract as compared to untreated extract (0.19–0. 31) and found lower ($p < 0.05$) as compared to vitamin C. The similar finding in trend was observed in the DPPH results. The results (DPFE3 > DPFE2 > DPFE1) indicate that reducing the power of DPFE can be enhanced with PEF application, which can convert the free radicals (FRs) to giving electrons for stable materials, because these FRs can strongly dismiss the reactions started [47].

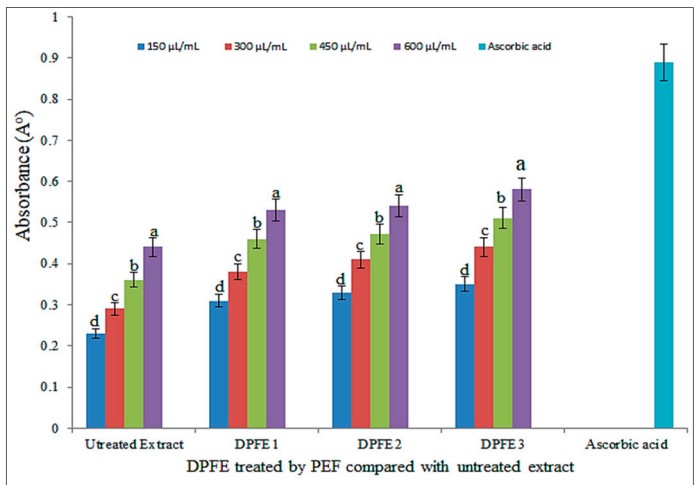

**Figure 2.** Reducing power assay of DPFE treated by PEF as compared to the untreated extract using ascorbic acid as reference. DPFE1: 1 kV/cm, DPFE2: 2 kV/cm, DPFE3: 3 kV/cm. The tests were performed as triplicates and values are mean ± standard deviation. Different superscript alphabets in a row show significant difference ($p < 0.05$) using Duncan's multiple range tests.

### 3.5. Color Parameters

Impact of PEF treatment on color attributes ($L^*$, $a^*$, and $b^*$), $h^o$ and $\Delta E$ is given in Table 3. The results were found similar to the findings reported previously by Rivas et al. [48]. However, their results did not show significant differences by application of high intensity of PEF to orange-carrot juice and date palm juice. The treatment by PEF with mentioned conditions parameters has no effect on color of DPFE. It proposes that PEF caused a little bit of change to the color of the food products treated by the methods and it was observed that effect on color was increased with the increase of voltage conditions [16,49]. Our results confirmed the previous findings of Mtaoua et al. [13], who reported a slightly significant variation with high-intensity (35 kV/cm using pulses of 4 µs pulses for 1000 µs at 100 Hz) to date palm (Bou-Hattem variety) fruits juice. Depending on the value of $\Delta E$, the color difference between all DPFE treated by PEF and untreated extract can be compared with previous findings of Barba, et al. [50], such as (6.0–12.0) great, (3.0–6.0) well visible, (1.5–3.0) noticeable, (0.5–1.5) slightly noticeable and (0–0.5) not noticeable. The present findings indicated that the color difference was found not noticeable in DPFE1 and DPFE2, while slightly noticeable in DPFE3.

**Table 3.** Color attributes of DPFE treated by PEF compared with the untreated extract

| Sample | Color Parameters | | | | | |
|---|---|---|---|---|---|---|
| | *L\** | *a\** | *b\** | *Hue (h°)* | *C\** | Δ**E** |
| Untreated extract | 33.40 ± 0.19 [a] | −0.98 ± 0.17 [a] | 4.25 ± 0.16 [b] | −85.66 ± 0.33 [b] | 4.36 ± 0.10 [b] | — |
| DPFE1 | 33.22 ± 0.13 [a] | −0.61 ± 0.11 [a] | 4.51 ± 0.18 [b] | −82.63 ± 0.29 [b] | 4.55 ± 0.11 [b] | 0.28 |
| DPFE2 | 33.12 ± 0.10 [a] | −0.58 ± 0.12 [a] | 4.70 ± 0.10 [b] | −81.93 ± 0.2[b] | 4.74 ± 0.20 [b] | 0.49 |
| DPFE3 | 32.96 ± 0.18 [b] | −0.29 ± 0.14 [b] | 5.00 ± 0.11 [a] | −73.21 ± 0.25 [a] | 5.08 ± 0.23 [a] | 1.33 |

DPFE1: 1 kV/cm, DPFE2: 2 kV/cm, DPFE3: 3 kV/cm, *C\**: Chroma value. The tests were performed as triplicates and values are mean ± standard deviation. Different superscript alphabets in a row show significant difference ($p < 0.05$) using Duncan's multiple range tests.

### 3.6. TSS, pH, TA, EC, and 5-HMF

The EC, pH, TSS, TA, and 5-HMF are presented in Table 4. According to the statistical analysis, there were no significant differences ($p < 0.05$) observed in TA, EC and pH in DPFE treated by PEF as compared to compared untreated extract; however, the values were found the highest in the treated extract. Overall, EC of liquid food is referred to the presence of some nutrient components such as macro and microelements, vitamins, amino acids and lipids [45]. Influence of PEF on TSS (°Brix) of DPFE is also reported in Table 4. This indicates that there was a non-significant impact ($p < 0.05$) among all DPFE treated by PEF as compared to untreated extract. The previous study shows 80% sucrose extractions within 1 h, when 7 kV/cm is applied, with 20 pulses and by changing the temperature from 40 °C to 70 °C [51]. For the results concerning HMF in DPFE treated by PEF, a significant increase ($p < 0.05$) in HMF was observed as compared to untreated extract with increase of electric field intensity, while no significant difference ($p < 0.05$) was observed in DPFE1 and DPFE2. The formation of HMF is associated with production of reactive carbonyl group from antioxidant molecules, which is mostly observed in non-enzymatic reaction [52].

**Table 4.** EC, TSS, pH, TA and HMF of DPFE treated by PEF compared with the untreated extract

| Parameter | Untreated Extract | Treated Extract | | |
|---|---|---|---|---|
| | | DPFE1 | DPFE2 | DPFE3 |
| EC (ms/cm) | 2.78 ± 0.11 [a] | 2.79 ± 0.11 [a] | 2.79 ± 0.11 [a] | 2.80 ± 0.11 [a] |
| TSS (°Brix) | 20.0 ± 0.09 [a] | 20.70 ± 0.03 [a] | 20.73 ± 0.10 [a] | 20.98 ± 0.11 [a] |
| pH | 5.85 ± 0.12 [a] | 5.79 ± 0.09 [a] | 5.78 ± 0.17 [a] | 5.76 ± 0.13 [a] |
| TA (%) | 0.10 ± 0.07 [a] | 0.11 ± 0.08 [a] | 0.11 ± 0.07 [a] | 0.12 ± 0.10 [a] |
| HMF (mg/L) | 4.95 ± 0.13 [c] | 5.11 ± 0.10 [b] | 5.13 ± 0.10 [b] | 5.94 ± 0.09 [a] |

EC: electrical conductivity, TSS: total soluble solids, TA: titratable acidity, HMF: hydroxymethyl furfural, DPFE: date palm fruit extract and PEF: pulse electric field. DPFE1: 1 kV/cm, DPFE2: 2 kV/cm, DPFE3: 3 kV/cm. The tests were performed as triplicates and values are a mean ± standard deviation. Different superscript alphabets in a row show significant difference ($p < 0.05$) using Duncan's multiple range tests.

### 3.7. Volatile Flavor Compounds

The numbers of the volatile flavor compounds in DPFE treated by PEF compared to untreated extract are presented in Table 5. Twenty-five compounds were identified in DPFE3 as the highest number of components, followed by DPFE2 and DPFE1 (24 and 23 components, respectively), compared with the number of components detected in the untreated extract (22 compounds). According to these results, pre-treatment by PEF has a positive impact on the aromatic compounds in DPFE and increase the total number of volatile components. It can also be seen in Table 5, compared with functional groups in all treated samples, acid compound groups, methyl, acids, ethyl and ester were found in higher proportions. Methyl, ethyl and esters are a significant aromatic group in aromatic compounds which lead to improve the sensory characteristics of the fruit drink [53,54]. Therefore, PEF can be applied to enhance the extractability of the volatile constituents. There were some components such as

9,12-Octadecadienoic acid (Z, Z) and n-hexadecanoic acid, that were identified in DPFE belonging to unsaturated fatty acids (Table 5). PEF assisted ethanolic extraction of bioactive components such as 5-HMF and n-hexadecanoic acid. These components were recommended as bioactive molecules due to their anti-inflammatory, antibacterial and antioxidant activity [55]. PEF might cause a significant variation in volatile compounds that could be due to an improved release of matrix-bound compounds. PEF treatment increased the intracellular contents extraction through permeabilization, resulting in more yield, improve extraction efficiency and intracellular metabolites extraction [10]. Notably, more volatile compounds with higher concentration were detected in the samples treated by PEF. Therefore, this treatment can be used to improve the quality of date palm fruits and their products such as wine, juice and vinegar.

**Table 5.** Volatile flavors compounds of DPFE treated by PEF compared with the untreated extract.

| Area (%) | | | | Constituent |
|---|---|---|---|---|
| Untreated Extract | DPFE1 | DPFE2 | DPFE3 | |
| 1.60 ± 0.02 [c] | 1.65 ± 0.01 [b] | 1.66 ± 0.03 [b] | 1.77 ± 0.01 [a] | Urea, *N*-butyl-*N*-nitroso- |
| 4.40 ± 0.04 [c] | 4.40 ± 0.00 [c] | 4.45 ± 0.03 [b] | 4.56 ± 0.00 [a] | Morpholine, 4-methyl-, 4-oxide |
| 5.14 ± 0.02 [d] | 5.28 ± 0.03 [c] | 5.35 ± 0.01 [b] | 5.44 ± 0.02 [a] | 1,3,5-Triazine-2,4,6-triamine |
| 9.45 ± 0.02 [c] | 9.46 ± 0.02 [c] | 9.55 ± 0.02 [b] | 9.63 ± 0.02 [a] | 4 H-Pyran-4-one, 2,3-dihydro-3,5-dihydroxy-6-methyl- |
| 1.65 ± 0.03 [b] | 1.66 ± 0.04 [b] | 1.66 ± 0.02 [b] | 1.77 ± 0.07 [a] | 2(3H)-Furanone, dihydro-4-hydroxy- |
| 3.90 ± 0.01 [c] | 3.93 ± 0.02 [c] | 4.05 ± 0.05 [b] | 4.18 ± 0.09 [a] | Dimethylamine, *N*-(neopentyloxy)- |
| 5.38 ± 0.04 [c] | 5.40 ± 0.03 [c] | 5.48 ± 0.03 [b] | 5.66 ± 0.02 [a] | Isosorbide Dinitrate |
| 27.26 ± 0.03 [c] | 27.30 ± 0.02 [b] | 27.31 ± 0.01 [b] | 28.07 ± 0.00 [a] | 5-Hydroxymethylfurfural |
| 5.41 ± 0.03 [c] | 5.40 ± 0.00 [c] | 5.49 ± 0.02 [b] | 6.09 ± 0.03 [a] | 1,2,3-Propanetriol, 1-acetate |
| 1.26 ± 0.00 [d] | 1.35 ± 0.02 [c] | 1.44 ± 0.04 [b] | 2.19 ± 0.05 [a] | 2-Methyl-1-isopropyl(dimethyl)silyloxypropane |
| 7.85 ± 0.01 [c] | 7.88 ± 0.01 [bc] | 7.91 ± 0.03 [a] | 8.09 ± 0.04 [a] | 4 H-Pyran-4-one, 2,3-dihydro-3,5-dihydroxy |
| 1.92 ± 0.08 [c] | 1.92 ± 0.02 [c] | 2.09 ± 0.01 [b] | 2.33 ± 0.08 [a] | 2-Methyl-3,4,5,6-tetrahydropyrazine |
| 3.16 ± 0.03 [c] | 3.20 ± 0.04 [bc] | 3.21 ± 0.02 [b] | 3.41 ± 0.02 [a] | 3-Methyl-3-buten-1-ol, TMS derivative |
| 3.57 ± 0.02 [c] | 3.63 ± 0.08 [b] | 3.85 ± 0.02 [a] | 3.89 ± 0.07 [a] | Glycoluril |
| 2.11 ± 0.04 [c] | 2.15 ± 0.09 [b] | 2.19 ± 0.05 [b] | 2.44 ± 0.06 [a] | Propanamide, *N,N*-dimethyl- |
| 4.07 ± 0.09 [c] | 4.07 ± 0.03 [c] | 4.15 ± 0.03 [b] | 4.36 ± 0.01 [a] | Butanoic acid, 2-methyl-, 2-methylpropyl ester |
| 2.20 ± 0.07 [d] | 2.27 ± 0.02 [c] | 2.39 ± 0.02 [b] | 2.75 ± 0.02 [a] | 1-Nitro-2-acetamido-1,2-dideoxy-ᴅ-glucitol |
| 1.01 ± 0.00 [d] | 1.09 ± 0.01 [c] | 1.18 ± 0.04 [b] | 1.55 ± 0.09 [a] | β-ᴅ-Glucopyranose, 4-*O*-β-D-galactopyranosyl- |
| 3.44 ± 0.01 [c] | 3.45 ± 0.00 [c] | 3.56 ± 0.09 [b] | 3.77 ± 0.07 [a] | 3-Deoxy-ᴅ-mannoic lactone |
| 2.68 ± 0.02 [d] | 2.68 ± 0.03 [c] | 2.80 ± 0.05 [b] | 3.09 ± 0.03 [b] | 3-Deoxy-ᴅ-mannoic lactone |
| 1.06 ± 0.00 [c] | 1.11 ± 0.01 [c] | 2.13 ± 0.04 [b] | 2.55 ± 0.09 [a] | n-Hexadecanoic acid |
| 1.50 ± 0.03 [d] | 1.61 ± 0.02 [c] | 1.88 ± 0.03 [b] | 2.13 ± 0.07 [a] | 9,12-Octadecadienoic acid (Z, Z)- |
| ND | ND | ND | 2.02 ± 0.02 [a] | 11,13-Dihydroxy-tetradec-5 ynoic acid, methyl ester |
| ND | ND | 0.59 ± 0.05 [b] | 0.88 ± 0.04 [a] | 2-Myristynoyl pantetheine |
| ND | 0.53 ± 0.04 [b] | 0.90 ± 0.01 [a] | 0.92 ± 0.03 [a] | Paromomycin |

DPFE1: 1 kV/cm, DPFE2: 2 kV/cm, DPFE3:3 kV/cm. The tests were performed as triplicates and values are mean ± standard deviation. Different superscript alphabets in a row show significant difference ($p < 0.05$) using Duncan's multiple range test. ND: Not detected.

## 4. Conclusions

Non-thermal PEF application is one of the prerequisite for sustainable processing due to innovative, low energy consumption and cost-effective in food and drinks products. In the current study, we have shown that PEF application enhances the extraction and biofunctionalities of date palm fruits juice. The DPFE by PEF has many advantages associated with TPC, TFC, TAC and TCC enhancement. The PEF slightly affected the overall appearance of treatment DPFE3 but there was no significant effect observed in color parameters. Physicochemical properties such as EC, pH, TA and TSS were not affected by PEF treatment, while a non-significant influence on the content of 5-HMFin DPFE3 compared to others treatments was observed. Thus, the PEF can help in improved extraction yield of bioactive compounds which have significant protection potential against the oxidation as demonstrated through DPPH and

reducing power assays. Thus, we conclude that PEF technology has great potential to preserve and retain the quality of fresh juices and their nutrition of values.

**Author Contributions:** A.D. and M.F.M. conceptualized the studies, M.H.A., M.U.N. and Z.A. designed the mythological parameter and validity, A.F. help informal analysis and statistical analysis, A.S., M.F.M. and N.A. prepared original draft, N.A., X.A.Z. M.K.I.K. and A.A.M. performed writing, revision and editing of final draft under supervision of X.A.Z. All authors reviewed and approved the manuscript draft.

**Funding:** The National Natural Science Foundation of China (grant #21576099), The S&T projects of Guangdong Province (grant# 2017B020207001 and 2015A030312001), as well as the 111 Project (grant #B17018) supported the grant of this research.

**Conflicts of Interest:** The authors declare no conflict of interest.

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
