# Peer review of "Pulsed Electric Field-Assisted Ethanolic Extraction of Date Palm Fruits: Bioactive Compounds, Antioxidant Activity and Physicochemical Properties"

_processes, doi:10.3390/pr7090585_

Round 1

Reviewer 1 Report

Ref: Processs-545619

Title: Pulsed electric field-assisted ethanolic extraction of date palm fruits: Bioactive compounds, antioxidant activity, and physicochemical properties

Thank you so much for giving me this chance to review this paper. In this paper, authors evaluated the pulsed electric field (PEF) on extraction  of bioactive components; antioxidant activity and physico-chemical properties of date palm fruit extract 19 (DPFE) compared to the extract untreated by PEF. Minor revision about this study are:

1. First page “article title” incorrect format.

2. The authors can further re-organize the abstract to be more concise and conclusive.

3. Introduction section should add description about “PDF” and is it currently used? If so, what plant extraction is applicable?

4. Suggested add “graphical abstracts”.

5. Line 177, 202, 230, 245, 263 and 282: suggested “ ND: Not detected “ delete.

Author Response

All queries, suggestion and comments have been addressed. please find in attached file herewith. 

Thanks 

Reviewer 2 Report

The methode used to improve the extraction is very important , however, the contains of the extracts remained unidentified. Mixtures of biomolecules, obtained by extractions are very interesting, but the use of these in medicine is more then questionable. It would be acceptable this work in the case of defined and identified molecules and structures. Moreover, the data of Table V can not be elucidated without the role, origin and structure identification of the mentioned compounds.

Author Response

The suggestions, comments and queries have been addressed in manuscript file and highlighted. Please find the file attached herewith. 

Thanks 

Reviewer 3 Report

Author documented this manuscript with detailed explanation of pulsed electric field-assisted ethanolic extraction of date palm fruits. However, there are some additional data needed. The comments are given below.

1. Author should go through the manuscript very carefully to fix numerous typos. For example, page 3, line 83, at ratio 0f. Author put zero instead of letter O. Also, the same line, 6 hrs should be written as 6 h. There should space between 45 and °C. etc., These mistakes are repeated several times in the manuscript.

2. Also, check the typos in table 2 title in page 6. 

3. It is recommended that author can say about characterization of PEF-induced in the cells of date palm fruits. Also, include the bar diagram of the cell disintegration index (Zp) of date palm fruits as a function of electric field strength (KV/cm) and total specific energy input.

Author Response

(The authors gave the same response as above.)

Round 2

Reviewer 2 Report

I have studied the answers of Authors and the revised manuscirpt. Now it is eligibler for publication.